# Development and Characterization of Fenugreek Protein-Based Edible Film

**DOI:** 10.3390/foods10091976

**Published:** 2021-08-24

**Authors:** Neha Kumari, Sneh Punia Bangar, Michal Petrů, R.A. Ilyas, Ajay Singh, Pradyuman Kumar

**Affiliations:** 1Department of Food Engineering and Technology, Sant Longowal Institute of Engineering and Technology, Longowal 148106, India; nehakumari27info@gmail.com; 2Department of Food, Nutrition and Packaging Sciences, Clemson University, Clemson, SC 29631, USA; 3Center for Nanomaterials, Advanced Technologies and Innovations, Technical University of Liberec, Studentská 2, 461 17 Liberec, Czech Republic; michal.petru@tul.cz; 4School of Chemical and Energy Engineering, Faculty of Engineering, Universitiy Teknologi Malaysia, Johor Bahru 81310, Johor, Malaysia; ahmadilyas@utm.my; 5Centre for Advanced Composite Materials (CACM), Universitiy Teknologi Malaysia, Johor Bahru 81310, Johor, Malaysia; 6Department of Food Technology, Mata Gujri College, Fatehgarh 140307, India; ajay3singh@gmail.com

**Keywords:** fenugreek protein concentrate, edible film, tensile strength, water vapor permeability, Fourier-transform infrared spectroscopy, X-ray diffraction

## Abstract

The present investigation studied the physicochemical, mechanical, structural, thermal, and morphological attributes of a novel edible film formed from fenugreek protein concentrate. Films were produced at different pH—9, 10, 11, and 12—and the effect of the pH on the films was studied. As the pH increased, tensile strength increased while water vapor absorption decreased, which is interrelated to the surface morphological properties; as the pH increased, the surface became smoother and compact without any cavities. The films produced were darker in color. Fenugreek protein films exhibited good thermal stability. Fourier transform infrared spectroscopy (FTIR) revealed the presence of strong bonding for the films made at alkaline pH. X-ray diffraction analysis (XRD) indicated the major structure of the film was amorphous. The study demonstrated that the fenugreek protein concentrate film has influential characteristics and can be used as an edible packaging film.

## 1. Introduction

The diverse consumption of plastic as a packaging material has led to multiple generations of waste streams. The large volume of plastic waste has created serious havoc on the environmental problem. In an endeavor to preserve the environs, most countries have started to bring down the consumption of one-use plastics in food packaging, thereby decreasing the price of controlling pollution [1]. One of the most accepted phenomena using traditional plastic is the transfer of potentially toxic and harmful components from the packaging plastic matrix towards the wrapped food and this is ascribed to the photo-oxidation reaction [2]. The trending global spotlight on bio-economy and health awareness has focused on the evolution of justifiable plastics that should be biodegradable, eatable, safe, thermally resilient, and mechanically strong [3,4]. Biological protection of the food product directly governs the product life-span, and hence it is very important [5]. Packaging executes a salient role in the prevention of food waste and in the achievement of establishing safety goals by creating a barrier between the environment and the food [6]. These edible plastics could be utilized in food product’s smart and active packaging materials [4,7]. Food-packaging innovations have helped to meet-up the emerging demands of the functional food market. The advancing concept of intelligent and active packaging technology provides a diverse innovatory way to extend lifespan, quality improvement, and safe consumption of food products. Waste minimization and the trending demand for green or sustainable packaging made using plant extract, which is either edible or biodegradable, are important for a clean environment and health longevity [4]. Natural polymer-based edible films are biodegradable and non-toxic that possess easy storage and safe transportation [8]. Amongst the biomaterial protein, edible films from various sources impressively expand environmental-friendly films due to their relative abundance and better film-forming capability [9]. Protein-based films are ideal for hydrophilic surfaces and provide oxygen and carbon dioxide barriers [10,11]. Structural specifications of protein-embedded edible films impart lots of functionalities such as intermolecular bonding [12]. Protein-based films are anticipated to have a good oxygen barrier. Recently investigated protein to have an edible film of biodegradable terms are peanut protein, casein, whey protein, gelatin, soy protein, gluten protein, corn zein, and mung-bean protein [13,14,15].

Fenugreek seeds are a recent choice and a well-suited source of protein under food enrichment concerns. The proportion of seed protein in fenugreek accounts for about 25 to 38%, which is composed of globulin (27.2%), prolamine (7.4%), glutelins (17.2%), and albumin protein (43.8%) [16]. Fenugreek seeds reside under leguminous crops and are edible, rich in protein and dietary fibers but low in fat proportion. The standard of protein composed among fenugreek seed is as good as soybean protein, moreover, lysine availability in them is almost competitive as like soybean protein [17].

Additionally, the fenugreek seeds are rich in bioactive compounds such as polyphenols and saponins [17]. Polyphenolic compounds from fenugreek seeds exhibit anti-diabetic properties. They hold powerful antioxidant properties which control high blood cholesterol, improves reduction in cell death, aging, and strengthens the immune system [18,19]. Compounds extracted from fenugreek have promising biological activities, including protection against cancer, allergies, bacteria, malaria, and viruses [20]. This necessitates the expansion in the utilization and preparation of value-added materials.

Thus, the use of fenugreek protein to prepare edible film introduces a sustainable way out for intensifying protein utilization, creating novel environmental-friendly and bio-based packaging. The potential applications of fenugreek protein-based edible film on food packaging were investigated owing to the non-existence of published literature on fenugreek protein concentrate used for the development of edible film as packaging means.

## 2. Materials and Methods

### 2.1. Raw Materials and Reagents

Fenugreek seeds of a particular variety, HSHM-57 (Hisar Sonali), were collected from Haryana Agricultural University, Hisar. The chemical reagents used in the present study were of analytical grade and were purchased from Sigma Aldrich, Delhi, India. The fenugreek seeds were manually cleaned, all the foreign particles were removed, and air-dried. Duly cleaned fenugreek seeds were milled to flour using an electrical mixer grinder. After grinding, the milled flour was passed through an 80 mesh sieve and stored for further analysis.

### 2.2. Protein Extraction

Protein extraction was executed using the standardized procedure [21] with some modifications. 10 g of fat-free fenugreek seed flour was taken and then disbanded in 200 mL deionized water; the pH of the solution was maintained at 12 with the aid of 2.5 N NaOH. Maintain this solution at 45 °C for 120 min in a water bath followed through centrifugation at 8000 rpm for 20 min. The supernatant was collected in a flask, and the remaining residue was discarded. The pH of the solution was maintained 2 by adding 2.5 N HCl; again, centrifugation was done at 8000 rpm for 20 min. The sediment was collected and neutralized, dried, and stored.

### 2.3. Casting of Edible Film from Fenugreek Protein Concentrates

Fenugreek protein films were prepared using a casting method adapted by Paglione et al. [22] with some modification. The film-forming solution was prepared by mixing 10 g of fenugreek protein powder in 100 mL distilled water, and 2.4 mL glycerol was used as a plasticizer (30 g glycerol/100 g). The solution was stirred for 20 min using a magnetic stirrer and then heated to 40 °C keeping in mind that the pH of the solution was monitor and maintained at 9, 10, 11, and 12 using 1M sodium hydroxide solution. Stir again for 30 min, and subjected to heating in a shaking water bath at 70 °C for 20 min. The film-forming solution was cooled to 30 to 35 °C which is then poured into Teflon-coated baking trays. The films were dried in a tray drier at 30 °C for 24 h, and films were designated as F9, F10, F11, and F12. The films were stored in desiccators until further analysis.

### 2.4. Physicochemical, Structural, Morphological, and Thermal Analysis

#### 2.4.1. Tensile Strength

The tensile test was conducted with the help of Texture Analyzer (Stable Micro Systems T2i, Surrey, UK) as stated by the method of Sukhija et al. [23] with some modifications. Initially, the film samples were sliced into strips (1.5 × 6 mm) before conducting the test. The specimens were pre-fixed to the movable jaws of the instrument with an initial distance of 50 mm and cross-head speed of 0.5 mm/s. Maximum tensile strength (MPa) was determined.

#### 2.4.2. Elongation at Break (%)

The calculation of elongation at break of fenugreek protein concentrate film was done by the method explained by Sukhija et al. [23]. The percentage change was divided by stretching the film sample at the moment of rupture by the initial length of the film gauge and then multiplied by 100. The measurements were repeated five times.

#### 2.4.3. Film Thickness

The thickness of the film sample was measured using a handheld micrometer manually (Mitutoyo 2046F) with an accuracy of ± 1 µm in 10 replicas for each film randomly. Mean values were calculated in mm.

#### 2.4.4. Water Vapor Permeability

The water vapor permeability (WVP) of the film sample was estimated by the modified gravimetric cup method, which is based on the method followed by Sharma and Singh [24]. Small lumps of anhydrous CaCl_2_ were used as a desiccant, and the mouth of the cups was sealed with a sealant. The cups were kept in a humidity chamber where the temperature-controlled environment was maintained at 30 °C with 70% relative humidity (RH). At regular time intervals, the weight measurements were recorded for about two hours, and the difference in weight of the cups was represented as a function of time. The slope was calculated according to the linear regression method. Water vapor transmission rate (WVTR) was estimated by dividing the slope of the straight line (g/s) by the area (m^2^) followed through permeation tests. 

#### 2.4.5. Transparency

The transparency of film was determined using a recording (UV–visible) spectrophotometer (Model UV160U-Shimadzu, Kyoto, Japan). The film strips were sliced in a rectangular configuration (2 mm × 8 mm). The filmstrips were fixed into the cuvette surface. For reference, an emptied cuvette was used. Transparencies of film-strips were estimated at 600 nm, and it was evaluated by the method of Han and Floros [25].

#### 2.4.6. Film Solubility

Following the method of Romero-Bastida et al. [26], the film samples’ solubility was evaluated. The weight of the pre-dried films cut-up into (20 mm × 20 mm) was taken. The pre-conditioned film strips were dipped in 50 mL distilled water. It was kept for one hour at ambient temperature, and after one hour, the insoluble films were separated carefully. The film samples, dried in a hot air oven at 60 °C till constant weight was observed.

#### 2.4.7. Moisture Content

The moisture content of the fenugreek films was determined as per the methods suggested by Sukhija et al. [23] wherein, the initial weight of the empty petri dish was recorded, and then pre-weighed film samples (2.0 g) were cut and placed in glass petri-dish. The film samples contained in petri-dish were transferred into the hot air oven at temperature 105 °C for 24 h. Moisture content was evaluated on the wet basis. 

#### 2.4.8. Color Measurements

Color measurement of fenugreek film samples was evaluated using the Hunter colorimeter fixed with an optical sensor (Hunter Associates Laboratory Inc., Reston, VA, USA) based on CIE *L**, a*, *b** color system. Where the *L** value indicates the lightness, its value ranges from 0 to 100, *a** value gives the degree of the red and green color, with a higher positive *a** value indicating more redness. The *b** value indicates the degree of yellow and blue color, thus the elevated *b** value signifies more yellow color. The instrument was standardized using white and black tiles before sample measurements.

#### 2.4.9. Fourier Transform Infrared (FTIR) Spectroscopy

The film samples were stored in a sealed desiccator with silica-gel (~0% RH) for 7 days at ambient temperature to get dehydrated film samples before FTIR analysis. PerkinElmer Spectrum-400 (U.S) was used for the FTIR analysis. The FTIR spectra of the film samples were recorded in the FTIR spectrophotometer between the range of wavenumbers from 4000 to 400 cm^−1^.

#### 2.4.10. X-Ray Diffraction (XRD)

An X-ray diffractometer (XRD) pattern of different films was obtained using (Pan analytical-Xpert PROMRD, X-ray diffractometer) with CoKa < 1 radiation. Each sample was canned in the range between 5 and 70° (2 h) at the rate of 1.20°/min (2 h) and with a step size of 0.05° (2 h).

#### 2.4.11. Scanning Electron Microscopy (SEM)

The dried film samples’ surface microstructure was examined by way of a Scanning Electron Microscope (JEOL, JSM 6300 SEM, JEOL, TOKYO; JAPAN). Film specimens were mounted on cylindrical aluminum stubs with the aid of double-sided carbon adhesive tape and sputtered with a thin layer of gold upon which the films were attached before the examination. 5 to 7 kV of acceleration voltage was used while testing the film specimens.

#### 2.4.12. Differential Scanning Calorimetry (DSC)

DSC measurements of the fenugreek protein films were conducted using a differential scanning calorimeter (Perklin-Elmer; DSC-4000; Mettler–Toledo; Columbus, OH, USA). Nitrogen gas was flushing (as cooling gas) at a rate of 20 mL/min. 3 to 4 mg of the fenugreek protein films were hermetically sealed into aluminum pans. The scanning temperature range of 20 to 200 °C at a rate of 10 °C/min was used. Peak temperature (Td) and denaturation enthalpy (∆H) data were collected for each film sample.

### 2.5. Statistical Analysis

For each experiment, triplicate runs were conducted, and the data collected were subjected to statistical analysis. StatSoft (Statistica 12.0) was used to determine data by analysis of variance (ANOVA). Duncan’s multiple range test was employed to evaluate significant (*p* ≤ 0.05) differences.

## 3. Results

### 3.1. Tensile Strength and Elongation at Break

Tensile strength may be defined as maximum tensile-stress sustaining capacity without failure by the sample while in operation under tension test [27]. It was observed that pH had a positive effect on the tensile strength of the fenugreek protein concentrate (FPC) embedded edible film indicating that the higher the pH, the higher the tensile strength was.

Due to protein-protein, electrostatic and hydrophobic interaction, protein films make complex structures, that govern the cohesiveness in films. pH plays a prominent role in protein films, developed as a result of water-soluble interaction. The solubilization, protein-denaturation, and unfolding are promoted in alkaline conditions; due to these interactions, the tensile strength increases. Table 1 summarizes the result of tensile strength and elongation at break. This tensile strength varied from 1.13 to 1.53 MPa, which is more than the result reported by Acquah et al. [28] for the film made from yellow pea protein concentrate, which was 0.18 MPa but less than film prepared from whey protein isolate which was 1.72 MPa. Zhao et al. [29] also found higher tensile strength for the film made at pH-12 than the film made at pH-11. He explained that more exposed protein structures induced increased reactive sulfhydryl groups, resulting in a considerable quantity of covalent bondings. Similarly, Bourtoom [14] and Sharma and Singh [24] found an increase in tensile strength with an increase in pH for mung-bean protein and sesame protein film. The same behavior was described by Cho et al. [30], who stated that the tensile strength of the film for pea protein had an increasing profile from 6.9 to 8.4 MPa with an increase in the pH 7 to pH 10.

The elongation at rupture may be defined as the measure of the film’s elasticity that refers to the maximum change in film length before breaking [31]. As verified for tensile strength, the linear effect of the pH significantly affected the elongation of the film; that is, with an increase in pH, the flexibility increased. Elongation at the break for the FPC film was found to be in the range of 6.74 to 14.87%. The same behavior was found by Cho et al. [30], who described that the elongation of the films behave variably instead of pH change and are much flexible at alkaline pH 12 for soy protein concentrate. All this is because of its treatment with alkaline pH, which may cause the rearrangement (dissociation and reaggregation) of soluble protein subunits. Zhao et al. [29] also observed the same profile with increasing pH; the elongation at break also increased. He explained that the protein is driven to a looser structure under higher alkaline conditions, increasing the film extensibility. pH significantly influenced the mechanical properties, concluding that higher pH values provide films with higher tensile strength and elongation. According to Rhim et al. [32], protein isolate films formed at alkaline conditions were more stable due to disulfide bonds, hydrophobic bonds and hydrogen bonds, indicating the contribution of covalent bonding in the stability of this type of films.

### 3.2. Water Vapor Permeability (WVP)

The water vapor permeability of edible films should be the lowest as possible. Polysaccharide-protein normally has higher water affinity and higher water vapor permeability when compared to synthetic polymers. Competitively, the WVP characteristic of protein-based edible films is low in comparison with polysaccharide films [14]. WVP in protein films depends on the hydrophilic nature of protein [33].

Table 1 summarizes the WVP of FPC films. The WVP of edible films from FPC ranged from 1.18 × 10^−10^ to 1.82 ×10^−10^ g.m/Pa.h.m^2^. The WVP of the films was less than lentil protein concentrate (3.09 × 10^−10^ g.m/Pa.h.m^2^) and soy protein isolate (2.6 × 10^−10^ g.m/Pa.h.m^2^) as reported by Bamdad et al. [34] and Soliman et al. [35], respectively. The effect of pH on the WVP of FPC edible films was observed to be decreasing with an increase in pH from 9 to 12. Similar results were observed by Zhao et al. [29] for chicken protein films, a decrease in WVP with the increase in pH from 11 to 12. Sharma and Singh [24] also reported decreasing profile in WVP with increasing pH for sesame protein isolate. Reduced WVP at elevated alkaline conditions (high pH) is due to protein solubilization, and protein unfolding which facilitates disulfide bondings by thiol oxidation reactions and thiol-disulfide interchange as reported by Shimada and Cheftel [36]. An increase in pH governs the denaturation of proteins, hence many expanded structures of proteins come into existence. These extended protein chains could be characterized through hydrogen bonding, ionic bonding, hydrophobic bonding, and covalent bonding. Increasing interactions could be observed ascribed to the homogeneous distribution of hydrophobic and polar groups. Resilient films with less WVP resulted due to increasing chain-chain interaction [37]. As the protein unfolding and denaturation start occurring, a strong protein-protein network is formed, and hence this is the reason for the lower water permeability of films. It can be concluded that FPC films had good water vapor barrier properties.

### 3.3. Film Thickness and Film Solubility

Thickness influences various attributes in edible film viz; mechanical properties, transparency, and water vapor permeability. A thin film leads to decreased mechanical properties and increased optical properties [38]. The thickness of the film specimen is affected by several factors, including composition, the amount of film-forming solution, and processing conditions. The barrier property of the film in terms of water vapor permeability attribute is in direct influence of film thickness, ascribed to the difference between the water vapor pressure and the moisture buildup onto the air-film interface. Film thickness and solubility extent are presented in Table 1. The thickness of the fenugreek protein concentrate film ranged from 0.23 to 0.30 mm which was increased as the pH of the film-forming solution raised from 9 to 12 pH. The reason could be the solubility and unfolding of the protein in the film-forming solution at higher pH. As the pH increased, the protein became more soluble, reducing the intermolecular space, and as a result, the thickness increased.

Solubility represents the applicability of film to pack foods rich in water and can be considered a significant factor in determining biodegradability. Usually, excess solubility indicates lower water defiance. It was observed that pH plays an important role in the solubility of edible films that somehow reduces the solubility level by increasing the pH. In other terms, the rise in pH of the film-forming solution showed an increase in the cohesiveness of film samples, hence condensed and strong films with low water solubility appeared. Alkaline conditions promoted increased interaction between the protein molecule and the protein cross-links resulting in reduced solubility [29]. Solubility of fenugreek protein edible films ranged from 49.05 to 68.55%. Saremnezhad et al. [39] reported the same for faba bean protein and observed decreased solubility with the increase in pH.

### 3.4. Moisture Content and Transparency of Edible Film Made from FPC

Moisture content determined the amount of moisture retained in the edible film sample. The high water resistance of the film is one of the most important properties for higher water activity of foods from a packaging perspective. Packaging materials that tend to engrossing moisture change the appearance, shelf life, and taste of the packaged foods [40]. Table 2 summarizes the moisture content and transparency.

The moisture content of FPC films ranged from 18.90 to 21.42%, and it decreased with an increase in pH. A previous study indicated a similar trend of decrease in moisture proportion with subsequent rise in pH terms [29]. Bamdad et al. [34] reported 23% moisture content for lentil protein film, which is more than fenugreek protein films. High moisture food samples required high water resistance packaging materials [41].

Transparency directly controls the consumer acceptability of films when applied for packaging or food coating [42]. Higher values indicate less transparency and more opacity. As reported by Mali et al., [43], the opacity of films also depends upon the thickness of films, with thicker films resulting in more opaque samples. A decrease in the transparency of films may reduce the exposure of food to light, which may alter the quality attribute of the food due to photocatalytic reactions, which may produce activated free radicals. Fat and oil oxidation, pigment discoloration, formation of off-taste, and vitamin A, B, and C loss may be reduced [44,45].

The FPC film had transparency in the range of 13.16 to 17.59%. As the pH of the film increased, the transparency increased. This may be because greater cross-linking and aggregation of protein make the film a solution more turbid, reducing transparency [29]. A similar result was reported by Sharma and Singh [24] for sesame protein film, while fenugreek protein film samples displayed lesser clarity than the whey-protein isolate films [46]. Lee et al. [47] reported the transparency value of composite film in the range of 14.35–45.53% made from nano clay and sesame protein.

### 3.5. Color of Edible Film Made from FPC

The color of the FPC films was evaluated by Hunter lab Colorimeter. Table 2 represents the color value of the films. Films showed *L** value in the range of 34.06 to 64.62, *a** value in the range of 5 to 14.79, and *b** value 2.25 to 18.13, which indicates the dark yellow color of the film. The film-forming solution mostly affected the color of the film. In contrast, heating temperature and heating time had less effect. Films made at less alkaline pH showed lighter yellow color than the films made at higher pH. Instrumental color parameter *L** decreased with an increase in pH and caused the films to appear darker. At alkaline pH, protein forms complexes with polyphenolic compounds. The complexes may have contributed to the discoloration of the films made at higher pH. Dark colors of the films were obtained, which might be used for packaging light-sensitive foods. Bourtoom [14] observed the dark mung bean protein films’ dark color at alkaline pH with *L** value 21.60 and *b** value 5.55. Choi and Han [48] reported *L** to value 38.8 for peanut protein film, while Sharma and Singh [23] observed *L** 44.33 *a** value 10.06 and *b** value 17.54 for sesame protein, similar to the fenugreek protein concentrate film. *L** values of protein from sesame and composite nano clay films were stated between the range of 48.59–62.24 [47].

### 3.6. FTIR Analysis

The infrared spectroscopy is determined based on radiation absorption and the investigation of multi atomic ions and vibration of molecules. Organic compounds are mainly identified by this method, and the type of functional group is determined [49]. FT-IR spectra of edible films made from fenugreek protein concentrate were analyzed, and its graph is illustrated in Figure 1. The intermolecular interactions were determined. As the pH of the film increased shift in bond lengths was observed. Strong bonds can be seen in films made at higher alkaline pH. The absorption peaks were mainly located in the spectral range of 900–1150 cm^−1^, characterized by bands of glycerol. 1200 to 1350 cm^−1^ associated with the combination of N–H in-plane bending and C–N stretching vibrations (amide III). 1400 to 1550 is attributed to N–H bending (amide II). 1600 to 1700 cm^−1^ is characterized by C=O stretching vibration and C–N vibration stretching vibration (amide I). 2850 to 2980 cm^−1^ is associated, with C–H stretching, and 3000 to 3600 cm^−1^ is attributed to free and bound O–H and N–H groups [50,51,52]. 1400–1550 cm^−1^ represents the amide II band due to N–H bending. Peaks in between 1400 and 1550 were associated with an amide I band governing the stretching vibrations of C=O and C–N groups. 2923 to 2928 cm^−1^ is characterized by Amide B, which is related to NH stretching vibrations. 3274.60 to 3374.90 cm^−1^ represents Amide A mainly ascribed to the stretching vibrations of N-H groups. It is completely limited to the NH group, and it is inconsiderate to the arrangement of the polypeptide backbone. Its frequency is dependent on the strength of the hydrogen bond [53]. Amide I and amide II are very sensitive to their confirmation, the two major bands of the protein infrared spectrum.

### 3.7. X-RAY Diffraction Edible Film Made from FPC

X-Ray Diffraction was performed to investigate the amorphous structure of the films made at different pH. Figure 2 exhibits the X-ray diffractograms of fenugreek protein concentrate film. The XRD spectra of films showed well-defined characteristic peaks. The intensity of the diffraction peak showed only one main crystalline reflection in all the films in a range of (19.50 to 21.17°), representing that most of the structure in the film formed at different pH was amorphous. The peaks observed displayed good compatibility between fenugreek protein concentrate and glycerol as the peaks were broader. A broader peak also indicates a partial or less crystalline structure. Martins et al. [54] also reported a similar diffracting peak (20°), constituting the amorphous character of films prepared from locust bean gum and carrageenan.

### 3.8. Surface Morphological Properties

Scanning electron microscopy (SEM), employed to reveal morphological attributes and interaction among edible film make-up constituents. It provides a better perception of water vapor permeability, optical property, and mechanical property. SEM may be used to evaluate useful information of the film substrate such as homogeneity of film, pores, cavities, clumps, layer structure cracks, level of dispersion of components in the matrix, and surface smoothness [55]. Structural surface morphology of the fenugreek edible films was captured by the scanning electron microscope (SEM) at magnification 1500× shown in Figure 3A–D. The film made at pH 9 (Figure 3A) was observed to have a rough, irregular surface. As the pH of the film increased, the film exhibited smoother surfaces with a lesser presence of pores and cavities. The film made at pH 12 (Figure 3D) showed smooth and compact structures, which relates to lower water vapor permeability and higher tensile strength due to compactness and proper solubilization of protein. A similar structure was observed by Han et al. [4] for the films made from soy protein isolate.

### 3.9. Differential Scanning Calorimetry (DSC)

A differential scanning was done to investigate the thermal behavior of the film samples. Thermally resistant films must withstand temperature fluctuations inside the packaged foods and outside the environment [56]. The range taken for thermal scanning was 20 to 200 °C at 10 °C/min. DSC thermograms for the film samples showed one endothermic peak. It can be observed that the peak denaturation temperature increased as the pH of the film-forming solution also increased. The minimum peak denaturation temperature was observed to be 142.48 °C for the film made at pH 9, and the maximum of 149.85 °C for the film made at pH 12. Denaturation temperature increased gradually for the films made at pH 9, 10, 11, 12, which were 142.48, 144.13, 148.46, and 149.85 °C, respectively, as displayed in Figure 4. Ramos et al. [46] reported the thermal denaturation temperature of whey protein isolate to be 152.0 °C which is more than the result obtained. Sharma and Singh [24] reported a thermal denaturation temperature value of 108 °C for sesame protein films less than fenugreek protein films. Fenugreek protein films showed moderate thermal denaturation, suggesting that they can be used for packaging materials.

## 4. Conclusions

Given study explored fenugreek protein as a biomaterial to develop edible films that can be used for food-packaging purposes and edible applications. Fenugreek protein films were properly developed, and the attributes essential for the potential use of the film were determined successfully. The effect of pH was studied in 4 different ranges viz; pH from 9 to 12. The tensile strength, water vapor permeability, and solubility aspect of the fenugreek protein amended films were affected through pH fluctuations. The properties of the fenugreek protein concentrate edible films were similar to other edible films developed from protein. The best fenugreek protein concentrate film was obtained at pH 12. Edible films from fenugreek protein concentrate displayed better mechanical properties. Thus it can be further used as packaging materials. Film’s color was dark due to the darker color of fenugreek protein at higher alkaline conditions. Present communication explores a new horizon for upcoming researchers wherein pH significantly addresses fenugreek protein concentrate film properties with better keeping characteristics. When compared to synthetic films, the mechanical properties and the optical properties needs to be improved. Further research can be explored to ameliorate the fenugreek protein’s optical properties, and mechanical properties concentrate edible films.

## Figures and Tables

**Figure 1 foods-10-01976-f001:**
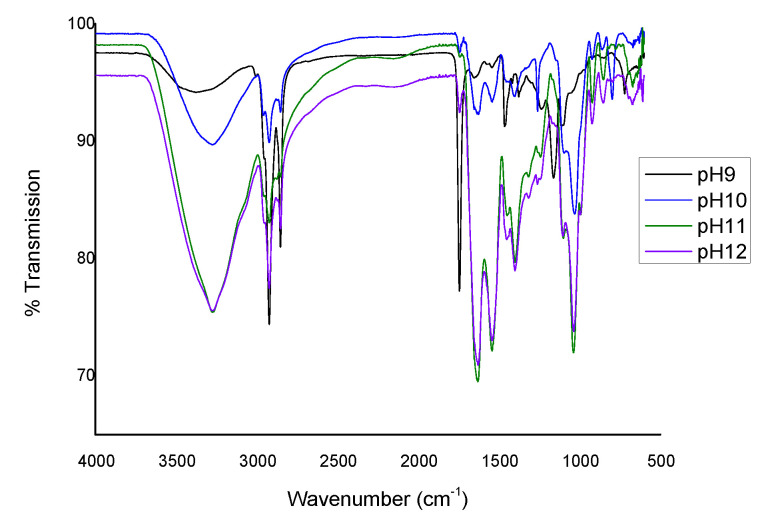
FTIR Spectrum of FPC edible film at pH 9, pH 10, pH 11, and pH 12.

**Figure 2 foods-10-01976-f002:**
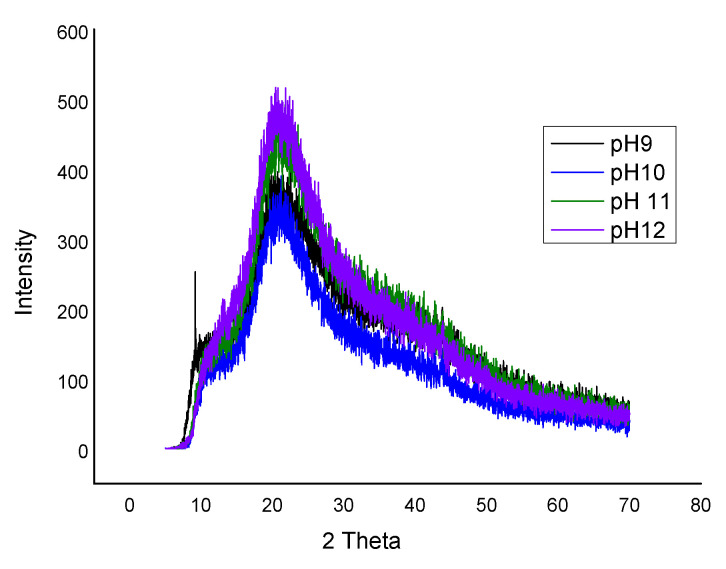
X-Ray Diffraction of FPC edible film made at pH 9, pH 10, pH 11, and pH 12.

**Figure 3 foods-10-01976-f003:**
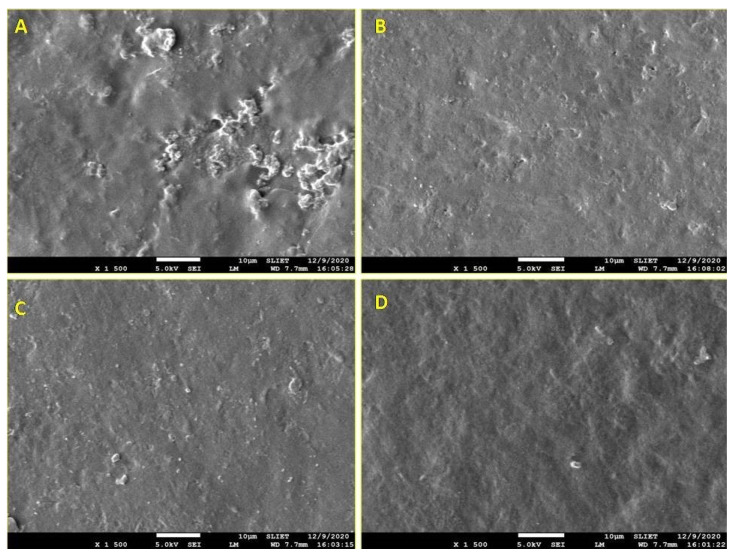
SEM image (×1500 magnification) of FPC edible film made at pH 9, pH 10, pH 11, and pH 12 shown as (**A**–**D**) respectively.

**Figure 4 foods-10-01976-f004:**
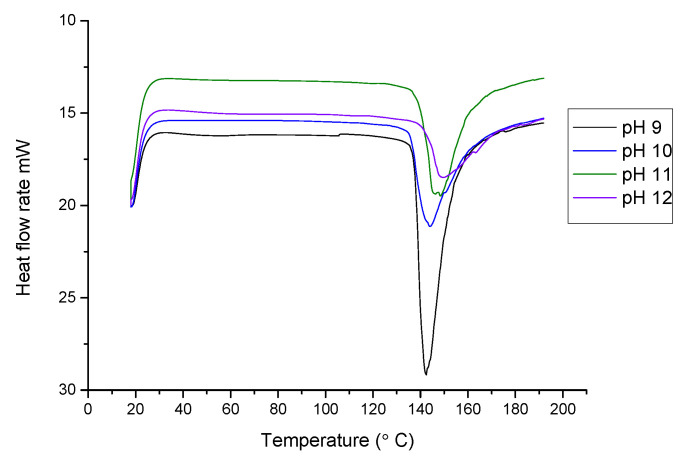
Differential scanning calorimetry (DSC) of FPC edible film made at pH 9, pH 10, pH 11, and pH 12.

**Table 1 foods-10-01976-t001:** Tensile strength, elongation at break, water vapor permeability, thickness, and film solubility of films prepared using FPC at different pH.

Film Samples	Tensile Strength (Mpa)	Elongation at Break (%)	Water Vapor Permeability (g.m/pa.h.m^2^) × 10^−10^	Thickness (mm)	Film Solubility (%)
F9	1.13 ± 0.16 ^b^	6.74 ± 0.12 ^d^	1.82 ± 0.11 ^a^	0.23 ± 0.10 ^a^	68.55 ± 0.21 ^a^
F10	1.32 ± 0.19 ^a,b^	7.67 ± 0.17 ^c^	1.60±0.13 ^b^	0.25 ± 0.12 ^a^	62.26 ± 0.23 ^b^
F11	1.49 ± 0.14 ^a^	11.74 ± 0.13 ^b^	1.40±0.16 ^c^	0.29 ± 0.11 ^a^	56.79 ± 0.19 ^c^
F12	1.53 ± 0.18 ^a^	14.87 ± 0.15 ^a^	1.18±0.12 ^d^	0.30 ± 0.12 ^a^	49.05 ± 0.20 ^d^

Mean ± SD values in a column followed by different letters a to d are significantly (*p ≤* 0.05) different. Results are means of triplicate determinations.

**Table 2 foods-10-01976-t002:** Moisture content, transparency, and color value of films prepared using FPC at different pH.

Film Sample	Moisture Content (%)	Transparency (%)	*L**	*a**	*b**
F9	21.42 ± 0.13 ^a^	13.16 ± 0.10 ^d^	64.62 ± 0.03 ^a^	8.39 ± 0.02 ^b^	18.13 ± 0.02 ^a^
F10	21.24 ± 0.10 ^b^	14.96 ± 0.13 ^c^	40.24 ± 0.01 ^b^	9.79 ± 0.01 ^a^	8.75 ± 0.03 ^b^
F11	21.24 ± 0.11 ^b^	16.78 ± 0.11 ^b^	37.20 ± 0.02 ^c^	5.89 ± 0.01 ^d^	2.25 ± 0.02 ^c^
F12	18.90 ± 0.12 ^c^	17.59 ± 0.13 ^a^	34.06 ± 0.01 ^d^	6.48 ± 0.02 ^c^	2.26 ± 0.01 ^c^

Mean ± SD values in a column followed by different letters a to d are significantly (*p ≤* 0.05) different. Results are means of triplicate determinations.

## Data Availability

Not Applicable.

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
