# Peer review of "Development and Characterization of Fenugreek Protein-Based Edible Film"

_foods, 2021, doi:10.3390/foods10091976_

Round 1
Reviewer 1 Report
Edible film studies are quite popular recently. The whole manuscript should be carefully edited because a lot of words are connected with a space. My comments are as follows.
- Page 1, line 2: characterization should be capitalized.
- Page 1, line 3: Dr. Punia is one of the co-corresponding authors, so please mark a star.
- Page 1, line 15: a space between Sneh and Punia.
- Page 1, line 20 and 22, a space between done. and As; line 22, a space between properties; and as.
- Page 1, lines 24-25: Please do not use abbreviations, FTIR and XRD, in the abstract as well as in the Keywords.
- Page 1, line 35: foodpackaging; line 36: themost; line 37: acceptedphenomenausing. Please correct.
- Page 2, line 54: [8].Amongst; line 55: expandenvironmental; line 57: carbondioxide; line 59: goodoxygen; line 64: foodenrichment. Please correct.
- Page 2, lines 67-69: these two sentences are related. How about combine them into one sentence?
- Page2, lines 70-71: ….polyphenols and saponins. This sentence requires a reference.
- Page 2, lines 77-78: there is no work….application. My personal minor suggestion, this sentence may be more suitable for next paragraph.
- Page 3, line 101: HCL should be replaced by HCl.
- Page 3, line 105: moderation?? or modification?
- Page4, line 162: (%) should be added in the formula.
- Page 5, line 171: ovenat; line 191: indicatesthe; line 205: diffractometer)with. Please correct.
- Page 5, line 179: the author placed the film samples in petri-dish and kept it at 105 degree Celsius. Is the petri-dish glass or plastic?
- Page 6, line 206: 5oto; line 220: wasused; line 234: playsa; line 235: inprotein; line 236: arepromoted. Please correct.
- Page 7, lines 263-264: ‘disulfide bond’ or ‘disulphide bond’, please consistent.
- Page 7-9, line 270: filmsevince; line 285: bonding,and; line 286: ascribedto; line 288: occurring,a; line289: forthe; line 342: [42],the; line 343: samples.A. Please correct.
- Pages 10-13, Fig. 1, 2 and 4: Minor suggestion, the authors can use F9-F-12 to replace pH9-pH12.
- Page 12, Figure 3: sub-figures cannot be connected as one figure; it will mislead readers. Also, the labels (A) to (D) is missing in the figure legend as well as in the main text. Please correct.
- Pages 12-13, line 443: temperatureincreased; line 455, (DSC)of; line 460: attributesessential; line 467: Thusit. Please correct.
Author Response
Respected Sir,
Your critical evaluation put this manuscript to very well side keeping in mind the journal's repo. all the suggestions are duly take care of and are amended with wise attention. submitted for your perusal and kind consideration, please.

Reviewer 2 Report
The main comments are as follows:
The manuscript gives light to the physicochemical, mechanical, structural, thermal, and 18 morphological attributes of a novel edible film formed from fenugreek protein concentrate. The large work sounds like interesting, however, there was less novelty. The results were just presented and described but not analyzed in a logical and deep way. Graphic abstract and highlights are necessary. And the language need to be improved.
Author Response
Respected Sir,
All the amendments as directed, are amended in green color alphabet. while working on to the revision for this draft, journal's repo are also in our mind hence submitting it for your perusal.
Regards;

Reviewer 3 Report
The use of novel protein sources as Component of Edible materials has been envisaged. I suggest the authors to remove the formulae from the materials and methods section. I also suggest to review in the introduction the paper by Corrado et al 2021 about the use of whey protein for preparing bioplastics.repared under alkaline pH as well. his paper was published in Food Hydrocolloids. However, the potential to perform biodegradabili test should at least be discussed.
Author Response
Respected Sir,
all the formulas were deleted from M&M section, new reference (as suggested) added to it. biodegradability test to these days is really critical owe to present Covid situation as we have no accessibility of lab.
Well, thanks to put all this framework to well refined and effective side through your guidance. Red Font color corrections are the responded part for our suggestions.
Regards;

Round 2
Reviewer 2 Report
The quality of the manuscript has been improved